# Understanding challenges to medical and dental student research practices. An insight from a cross-sectional study of the public sector in Pakistan

**Wajiha Qamar** *

Department of Oral Biology, Bacha Khan College of Dentistry Mardan, Mardan, Pakistan

* wajihaqamar.ob@gmail.com

**Data Availability Statement:** The data is available in excel sheet.

**Funding:** The authors received no specific funding for this work.

## Abstract

### Objective

The objective of the study is to identify and comprehend the challenges experienced by the undergraduate medical and dental students in enrolled in public sector in Pakistan while conducting research.

### Methodology

A cross-sectional study was carried out from April to June of 2023 among undergraduate students at public sector medical and dental schools in Khyber Pakhtunkhwa, Pakistan. A customized questionnaire was developed to gather information on the challenges faced by students when undertaking research. The survey was circulated online using Google Forms, and participation was entirely optional. Descriptive statistics were used to analyse the responses.

### Results

Participants in the study were 58 male and 139 female students who were enrolled in the Bachelor of Medicine & Bachelor of Surgery (MBBS) and Bachelor of Dental Surgery (BDS) programs. The analysis found that 47% of students cited a lack of knowledge and research skills as a major barrier, while 40% cited time restrictions as a major issue. Mentorship and training issues were cited as major barriers by 51% of students, while a lack of institutional support was cited as a significant issue by 53% of students. Language issues made it difficult to produce research papers for 14% of students, while finding research opportunities was challenging for 38% of students.

### Conclusion

According to the research, undergraduate medical and dentistry students in the public sector encountered a number of challenges when conducting research. It was suggested that these issues be resolved by include research projects in the curriculum, providing specific

**Competing interests:** The authors have declared that no competing interests exist.

interventions for enhancing research skills, establishing mentorship programs, and allocating funds for research activities.

## Introduction

In the rapidly evolving fields of medicine and dentistry, research fosters innovation and enhances patient care. It supports a person's professional advancement and personal development. It is impossible to overstate the significance of research to the education and growth of undergraduate medical and dental students. Research provides learners with knowledge about scientific inquiry, critical thinking, and problem-solving, which ultimately impacts how they see evidence-based practice [1]. Research can prove invaluable for undergraduate medical and dental students as it strengthens their abilities for critical thinking and problem-solving, enabling them to address challenging medical and dental conditions logically based on the best available information [2]. It fosters continuous education, develops a greater understanding of scientific concepts, and ensures students are informed about the most recent advancements in their respective fields. Another skill that researches experience provides learners with is the capacity to comprehend and critically analyse scientific material, empowering them to make informed clinical decisions in the future [3]. A study published in Journal of Dental Education discovered that dental students' critical thinking skills were higher to those of their peers when they engaged in research [4]. In addition to fostering a culture of lifelong learning, undergraduate medical and dental students who are actively engaged in research are exposed to the most recent scientific literature and methodologies, enabling them to stay up with the rapidly evolving body of knowledge. This approach of continuous education ensures that students have the skills to offer the highest level of care to their future patients. By doing more in-depth research on a subject, students can discover their passion and acquire specialized knowledge. This could prompt individuals to pursue further study or careers in specialized disciplines like academic research, dental specialty, or the various subspecialties of medicine. Additionally, research experience is highly valued by academic institutions and residency programs, offering students a competitive edge when applying for postgraduate studies or research careers.

The breadth of undergraduate research in Pakistan is expanding swiftly, providing students with several opportunities to participate part in research initiatives and help the medical field [5]. The establishment of research units within medical and dental colleges, which support students' research endeavours, serve as mentors for them, and encourage a research-cantered mindset, are merely a few of the platforms that have been established to support and promote the country's undergraduate medical and dental students in their research endeavours [6]. Few medical and dental colleges mandate research projects as a part of the graduation requirements, which encourages students to actively participate in scientific research. Additionally, universities and professional organizations have established institutional ethics review boards to further support research activities and ensure that ethical issues are included in student research projects. This safeguards the rights and wellbeing of research participants and ensures that research is carried out in conformity with international ethical standards. In Pakistan, it has been shown that the promotion of undergraduate medical and dental research and the development of a research culture are more pronounced in the private than in the public sector medical and dental colleges [7]. Despite of an enabling environment, research has been demonstrated to be poorly utilized, particularly in the public sector [7]. Therefore, it is imperative to comprehend the challenges that medical and dental students encounter when doing

research and publishing their findings in the public sector medical and dental colleges. This cross-sectional study was carried out with an objective to get a greater understanding of these issues, to offer a rationale for understanding the existing status of research use, and to identify areas for development. The study will not only identify the various factors that contribute to inadequate application of research findings in the public sector but also will serve as a helpful resource for the policy makers, academic institutions, and healthcare professionals as they develop strategies to address these challenges.

## Materials and methods

A cross-sectional survey was conducted from April to June 2023 to ascertain the challenges faced by undergraduate students of public sector medical and dental colleges in Khyber Pakhtunkhwa, Pakistan. The objective of this study was to discover about the difficulties that students encountered when conducting research. The survey included all academic years for undergraduate medical and dental students from public sector colleges. A customized questionnaire was developed in order to gather data. A senior faculty member was selected from each public medical and dental college in the province, and a WhatsApp group was established where these members were added and used as a means of contact with all of these faculty members. The questionnaire was uploaded on Google Forms, and the faculty member received the link through a WhatsApp group. The faculty member was tasked to distribute the link with the students either personally visiting the classes of all five academic years of MBSS students and four years of BDS students, or via WhatsApp groups for students. To achieve the greatest possible participation, two reminders were sent via the faculty, to the students.

The study was approved by Gandhara University ethics board, and was conducted with complete compliance to ethical standards. When conducting the study, ethical considerations were considered. Among other things, informed consent was acquired before respondents filled out the questionnaire, and it was made clear that the respondents' identities would be kept confidential, and that the data would be used as group data presentation for publication, and policy decisions to improve the research environment in undergraduate medical and dental institutions. The research's objectives and purpose were explained to the students, and it was underlined that their participation was completely voluntary. Students who were willing to engage in the study and were enrolled in public sector medical and dentistry colleges were deemed eligible for the research.

Descriptive statistics were utilized to summarize the responses and provide a broad perspective of the challenges faced by undergraduate medical and dental students in research and publication.

## Results

According to the results, overall 139 females and 58 males took part in the study. The study included data on the distribution of individuals by age and gender. The participants were enrolled in the Bachelor of Dental Surgery (BDS) and the Bachelor of Medicine, Bachelor of Surgery (MBBS), two distinct academic programs, public sector of Pakistan's Khyber Pakhtunkhwa province. The responders consisted of 108 female BDS students and 26 male BDS students. Similarly, 31 students from the MBBS program and 32 male students in total participated in the study.

Our study found, that 40% of students mentioned the time limitations as a serious issue and one of the biggest impediments. Lack of knowledge and research abilities was another significant problem that 47% of students identified as a major barrier. Mentorship and training obstacles were significant hurdles, as seen by the 51% of students who assessed them as

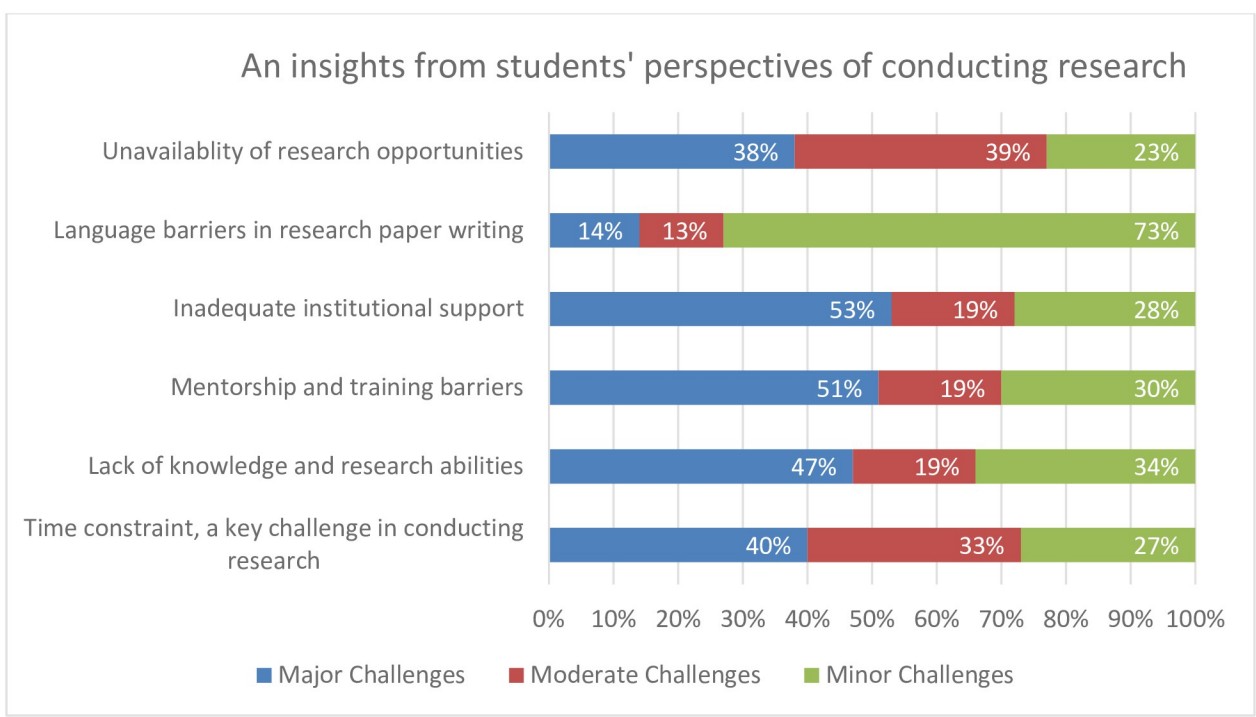

**Fig 1. Public sector undergraduate students' perspectives of conducting research.**

substantial issues. Furthermore, poor institutional support was listed as a key issue by 53% of students. 14% of students said that writing research papers was difficult because of language barriers. Finding research opportunities was very challenging for 38% of students. The Fig 1 gives an overview of public sector undergraduate students perspectives of conducting research.

## Discussion

The present study provides significant insight into the challenges that impede undergraduate students in the public sector who are studying medicine and dentistry and striving to conduct research. Time is critical in undergraduate medical research because of limited availability based on by clinical rotations, assignments, and academic obligations. Setting aside adequate time for research tasks such as literature reviews, data collection, analysis, and writing can be challenging for students.

Effective research participation is hampered by this restriction, which also limits the investigation of prospects and impedes academic and professional development. Time constraints also surfaced as a significant problem, with 40% of overall students citing them as a very serious obstacle. The results is consistent with a similar study conducted in Pakistan, which discovered that time constraints restricted junior medical faculty in four medical universities/ teaching hospitals from conducting research [8]. Similar studies conducted in neighbouring countries, such Saudi Arabia have also found that time constraints pose a barrier for medical students [9].

The findings of our research emphasize the need of effectively utilizing time-management techniques and including research activities into the curriculum to address this persistent issue. Lack of knowledge and research skills was mentioned as another significant obstacle by

47% of students. The result aligns with studies conducted in Pakistan and the surrounding regions [8, 10]. Similarly, another study found that medical students perceived themselves as lacking the necessary research skills [11]. Our study adds to the body of knowledge in this field by highlighting the problem of poor research knowledge and skills among medical and dentistry students and the requirement for targeted interventions to enhance research capabilities.

Mentors plays a crucial part in undergraduate research by providing students with assistance, encouragement, and relevant learning experiences. A mentor is a skilled advisor who may aid students in developing their research skills, guiding them through the research process, and fostering both their personal and professional growth. Effective mentoring has been shown to help many aspects of undergraduate research, including research productivity, skill development, self-confidence, and career objectives. Our study found that 51% of students considered mentorship and a lack of capacity building to be significant barriers. This finding is consistent with those of earlier research that was done in the area. For instance, a study conducted in Pakistan by Sabzwari et al. found that medical students had difficulty locating qualified mentors and obtaining support throughout the research process [8]. Similarly, another study found that medical students' inability to find mentors is a challenge to their participation in research [12]. It is advisable that academic institutions should set up effective mentorship programs and offer structured training opportunities to encourage mentorship and support for research in order to address this issue.

An enabling environment, which includes institutional resources, guidance, and mentoring to improve productivity, skill development, and research involvement, is necessary for undergraduate medical and dentistry students to conduct research [13]. Our study found that 53% of students mentioned poor institutional assistance as a major barrier impeding research culture. A review of primary and secondary sources and interviews with prominent scholars from seven higher education institutions in Pakistan found the lack of institutional support as a significant hurdle [14]. Additionally, another study found that a lack of institutional support made it difficult for medical students to participate in research [3, 15]. These findings emphasize the significance for establishing a supportive environment for research within academic institutions, including the provision of resources, facilities, and funding for research.

Undergraduate students need to be skilled in research in order to conceptualize research, design research, analyse data, and write effectively. 14% of our respondents specified that they had trouble writing research papers. Although there is little research on this particular topic in the area, but a study conducted in six Arab countries found that language issues were a barrier to medical students participating in research [16]. This study points to the need for providing an enabling environment by identifying the resources that are specifically designed to enhance the capacities of the students overcome these challenges and advance their research writing abilities.

## Conclusion

Our research has found that public institution undergraduate medical and dental students encounter a range of impediments when undertaking research.

1. Time constraints, clinical rotations, coursework, academic obligations, and other commitments make it difficult for students to dedicate time to research activities. Research projects must be incorporated into the curriculum in order for it to be sustainable and institutionalized. By using effective time management techniques, decision-makers can address these problems and ensure that students have enough time to participate in research activities.

2. Inadequate knowledge and expertise necessitate specialized interventions to improve their research skills. Programs like research workshops, seminars, and courses that emphasize critical analysis, scientific writing, and research methodology may be launched by decision-makers. Professors can assist students in overcoming this obstacle and actively participating in research by giving them access to appropriate resources.

3. Mentorship and training encountered several difficulties. Official mentorship programs can be established linking students with knowledgeable mentors who are able to assist them at each step of the research process. Faculty can help students develop their research skills as well as their personal and professional growth by providing them with access to training opportunities including research technique seminars and conferences.

4. Decision-makers in public sector institutions should set aside resources particularly for research activities. In addition to sponsoring research grants and scholarships, this entails granting access to research databases, labs, and tools. Undergraduate students will be inspired to actively participate in research activities if a friendly environment is developed.

## Limitations

The study's lack of funding had an effect on its methodology and sampling approach. The following are a few limitations

1. There is a significant risk of limited accessibility and participation as a result of the survey's distribution through faculty members and class representatives, which could result in biased sample selection and unequal representation.

2. The dependence on online platforms and WhatsApp groups, which can exclude those without internet access or those who are not subscribers to the given communication channels, further restricts the generalizability of the findings.

3. Another issue is the cross-sectional survey methodology, which just provides a snapshot and makes it difficult to determine causality or changes over time.

## Supporting information

**S1 Data.**
(XLSX)

## Author Contributions

**Conceptualization:** Wajiha Qamar.

**Data curation:** Wajiha Qamar.

**Formal analysis:** Wajiha Qamar.

**Methodology:** Wajiha Qamar.

**Resources:** Wajiha Qamar.

**Validation:** Wajiha Qamar.

**Visualization:** Wajiha Qamar.

**Writing – original draft:** Wajiha Qamar.

**Writing – review & editing:** Wajiha Qamar.

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
