## [Decision Letter · Decision Letter 0]

12 Sep 2023

PONE-D-23-18552Understanding challenges to medical and dental student research practices. An insight from a cross-sectional study of the public sector in PakistanPLOS ONE

Dear Dr. Wajiha,

Thank you for submitting your manuscript to PLOS ONE. After careful consideration, we feel that it has merit but does not fully meet PLOS ONE’s publication criteria as it currently stands. Therefore, we invite you to submit a revised version of the manuscript that addresses the points raised during the review process.

Please submit your revised manuscript Oct 27 2023 11:59PM. If you will need more time than this to complete your revisions, please reply to this message or contact the journal office at plosone@plos.org. Please include the following items when submitting your revised manuscript:A rebuttal letter that responds to each point raised by the academic editor and reviewer(s). You should upload this letter as a separate file labeled 'Response to Reviewers'.A marked-up copy of your manuscript that highlights changes made to the original version. You should upload this as a separate file labeled 'Revised Manuscript with Track Changes'.An unmarked version of your revised paper without tracked changes. You should upload this as a separate file labeled 'Manuscript'.

We look forward to receiving your revised manuscript.

Kind regards,

Sally Mohammed Farghaly

Academic Editor

PLOS ONE

6. Please amend your authorship list in your manuscript file to include author Wajiha Qamar.

Reviewers' comments:

Reviewer's Responses to Questions

**Comments to the Author**

1. Is the manuscript technically sound, and do the data support the conclusions?

Reviewer #1: No

Reviewer #2: Partly

Reviewer #3: Partly

2. Has the statistical analysis been performed appropriately and rigorously? 

Reviewer #1: No

Reviewer #2: No

Reviewer #3: Yes

3. Have the authors made all data underlying the findings in their manuscript fully available?

Reviewer #1: Yes

Reviewer #2: No

Reviewer #3: No

4. Is the manuscript presented in an intelligible fashion and written in standard English?

Reviewer #1: Yes

Reviewer #2: Yes

Reviewer #3: No

5. Review Comments to the Author

Reviewer #1: This is an interesting study exploring challenges that keep medical and dental students from public colleges in Pakistan from participating in research opportunities. While the authors raise interesting and important questions, the methods and results of this study are very limited and need significant additional work before the manuscript would be ready for consideration for publication.

Methods:

Authors used a sampling frame of public colleges in a particular region. How many colleges in the catchment area? Public? Private? How many students at those colleges? Those numbers are needed to make sense of the participation.

Need at least 1-2 sentences describing how medical and dental public colleges work in Pakistan – number of years of study; years where medical/dental curriculum is primary, etc.

Survey development – what was covered in the survey? How many questions?

Statistical analysis – was analysis done in excel? Google sheets? Other methods? Need to specify.

Results:

There is very little analysis. Even knowing what we know about the survey authors should explore gender differences in responses, differences in year of school, etc. Authors should report participation rates from each school, and give some sense of what proportion of eligible students at each school participated. All of this is necessary to understand the context and to reliably interpret the results of this work.

Writing is rough in parts, and will need additional editing. A few editing suggestions:

L3 ‘dental students enrolled in public sector schools’

L20 ‘including research projects’

L49 ‘participate in research initiatives’

L51 ‘research-centered mindset’

L54 ‘which would encourage students’

L61 ‘Despite an enabling environment’ [here also need to clarify what kinds of research opportunities are available to public college students]

L67 ‘opportunities, and to identify areas for development.’ - I think you mean research opportunities, which is distinct from how someone uses the results of research.

L69 ‘for policy makers’

L73 is Khyber Pakhutunkhwa a Province?

L74 ‘discover the difficulties’

L76 ‘public sector colleges in Pakistan.’ Or ‘public sector colleges in Khyber Pakhutunkhwa.’

L81 ‘the WhatsApp group.’

L86-87 ‘ethical standards. Informed consent was acquired.’

L89 ‘group data in presentation for publication’

L93 ‘dentistry colleges in Khyber Pakhutunkhwa’

Reviewer #2: The topic has been explored in other studies, and the study lack rigor in terms of correlation between probable factors impeding research. Recommendations have been made part of the conclusion. The research questionnaire has not been shared. How was the sample stratified? Was there any difference between BDS and MBBS students?

Reviewer #3: Topic seems interesting but not any significant contribution is reflected.

1- Abstract need to be revised and main objective, method and major findings should be highlited.

2- LR is quite weak, need to add recent and latest contextual studies.

3- Methodology need to justify, why this sample is used? why this analysis approach is adopted?

4- Findings need to be reported in a systematic way.

5- Discussion is not much clear, it is suggested to support your argument with previous studies.

6- Implication of study need to be highlight.

Good Luck

6. PLOS authors have the option to publish the peer review history of their article (what does this mean?). If published, this will include your full peer review and any attached files.

Reviewer #1: No

Reviewer #2: No

Reviewer #3: No

---

## [Author Response · Author response to Decision Letter 0]

12 Sep 2023

I sincerely appreciate the valuable input from the reviewers and the editorial team, which has undoubtedly enhanced the overall quality of our work. I have modified the manuscript in the light of suggestion and provided a response matrix for the feedback provided.

Thank you once again.

Best regards

---

## [Decision Letter · Decision Letter 1]

24 Nov 2023

Understanding challenges to medical and dental student research practices. An insight from a cross-sectional study of the public sector in Pakistan

PONE-D-23-18552R1

Dear Dr. Wajiha,

We’re pleased to inform you that your manuscript has been judged scientifically suitable for publication and will be formally accepted for publication once it meets all outstanding technical requirements.

Kind regards,

Sally Mohammed Farghaly

Academic Editor

PLOS ONE

Additional Editor Comments (optional):

Reviewers' comments:

Reviewer's Responses to Questions

**Comments to the Author**

1. If the authors have adequately addressed your comments raised in a previous round of review and you feel that this manuscript is now acceptable for publication, you may indicate that here to bypass the “Comments to the Author” section, enter your conflict of interest statement in the “Confidential to Editor” section, and submit your "Accept" recommendation.

Reviewer #2: All comments have been addressed

Reviewer #4: All comments have been addressed

2. Is the manuscript technically sound, and do the data support the conclusions?

Reviewer #2: Yes

Reviewer #4: Yes

3. Has the statistical analysis been performed appropriately and rigorously? 

Reviewer #2: Yes

Reviewer #4: I Don't Know

4. Have the authors made all data underlying the findings in their manuscript fully available?

Reviewer #2: Yes

Reviewer #4: Yes

5. Is the manuscript presented in an intelligible fashion and written in standard English?

Reviewer #2: (No Response)

Reviewer #4: Yes

6. Review Comments to the Author

Reviewer #2: (No Response)

Reviewer #4: The author in the manuscript "Understanding challenges to medical and dental student research practices. An insight

from a cross-sectional study of the public sector in Pakistan" have addressed the reviewer's comments nicely and revised the whole manuscript accordingly. In the view of all the changes and addressing the comments I recommend the manuscript for publication.

7. PLOS authors have the option to publish the peer review history of their article (what does this mean?). If published, this will include your full peer review and any attached files.

Reviewer #2: No

Reviewer #4: **Yes: **Prof. Dr. Syed Ali Raza Naqvi

---

## [Editor Report · Acceptance letter]

6 Dec 2023

PONE-D-23-18552R1 

Understanding challenges to medical and dental student research practices. An insight from a cross-sectional study of the public sector in Pakistan 

Dear Dr. Qamar:

I'm pleased to inform you that your manuscript has been deemed suitable for publication in PLOS ONE. Congratulations! Your manuscript is now with our production department. 

Kind regards, 

on behalf of

Professor Sally Mohammed Farghaly 

Academic Editor

PLOS ONE